# Thidiazuron Promoted Microspore Embryogenesis and Plant Regeneration in Curly Kale (*Brassica oleracea* L. convar. *acephala* var. *sabellica*)

**Jiaqi Zou †, Xiao Zou †, Zhichao Gong, Gengxing Song, Jie Ren * and Hui Feng ***

College of Horticulture, Shenyang Agricultural University, No. 120 Dongling Road, Shenhe District, Shenyang 110866, China
* Correspondence: 2019500023@syau.edu.cn (J.R.); fenghuiaaa@syau.edu.cn (H.F.)
† These authors contributed equally to this work.

**Abstract:** Curly kale (*Brassica oleracea* L. convar. *acephala* var. *sabellica*), the most common type of edible kale, characterized by providing rich nutrition and health care functions, is sought after and has been listed as top of the healthiest vegetables in recent trends, and has aroused the interest of breeders in cultivating new varieties. However, it usually takes more than six years to obtain a homozygous kale inbred line for commercial seed production through conventional breeding procedures due to its long growth and development period. The isolated microspore culture (IMC) technique could be a time-saving alternative method for producing doubled haploid (DH) lines that are genetically homozygous. In this study, we successfully utilize the efficient cytokinin thidiazuron (TDZ) to promote microspore embryogenesis and plant regeneration in two curly kale cultivars ('Winterbor $F_2$' and 'Starbor $F_2$'). Compared with the control (0 mg/L TDZ), all tested TDZ concentrations (0.1, 0.2, 0.3, 0.4 mg/L) had no adverse effects on embryogenesis, and 0.2 mg/L TDZ had an optimal effect on embryo survival and plant regeneration of the two genotypes. For 'Starbor $F_2$', 0.2 mg/L TDZ treatment achieved the highest embryogenesis rate (1.83-fold higher than the control group) and direct seeding rate (1.61-fold increase), and the lowest mortality rate. Likewise, 0.2 mg/L TDZ increased the embryogenesis rate of 'Winterbor $F_2$' by 1.62 times, the direct seeding rate by 1.61 times, and the mortality rate fell to the lowest. A 1/2 Murashige and Skoog (MS) medium with 0.2 mg/L 1-Naphthaleneacetic acid (NAA) can significantly promote the rooting of the regenerated seedlings. These results provide new insights into the practical application of the IMC technique in shortening the breeding cycle of kale.

**Keywords:** curly kale; microspore culture; thidiazuron; embryogenesis; doubled haploid lines

## 1. Introduction

Kale (*Brassica oleracea* var. *acephala*) belongs to the Brassicaceae family with both edible and ornamental properties originating in the Mediterranean area. Edible leaf-vegetable kale represented by the curly kale variety, with dark green and curled leaves, has a long planting history in Europe [1]. Because of its wide adaptability, high healthcare value, and cold tolerance, the cultivation and consumption of edible kale varieties are greatly expanding all over the world [2]. Relevant studies have demonstrated that edible-type kale possesses a high nutritional value, rich in nutritional components, such as protein, fiber, antioxidants, calcium, and vitamins C and K [3–5]. Because it contains more vitamins and dietary fiber than *Brassica* and other vegetables, it has gained great popularity as a 'superfood' [2]. Even more to the point, whether in salads or smoothies, juicing, dried chips, or stews, the culinary uses of edible kale are almost endless [6]. The worldwide popularity of edible leafy kale has led to a surge in demand, prompting breeders to develop landraces suitable for local specific conditions [2]. Thus, the breeding of improved varieties and the cultivation of new varieties are of great significance to edible leafy kale.

Currently, most commercial ornamental kale varieties are $F_1$ cultivars, while most edible kale varieties are conventional varieties and a few hybrids, which lack the genetic diversity to meet the increasing demands [7]. As a biennial plant with strong heterosis, kale parent materials require multiple generations of geitonogamy to obtain pure inbred lines for approximately 6 years due to its self-incompatibility, and then it usually takes 2 to 3 years for comparative tests of the created hybrid combinations to be made. Consequently, the whole breeding cycle lasts 8 years. Despite conventional efforts having made great progress, they are still limited by the long breeding cycle [8]. Therefore, it is urgent to eliminate the lengthy inbreeding cycle, decrease the heavy workload, and speed up the progress of kale variety improvement.

Isolated microspore culture (IMC) technology is the primary approach most commonly used for rapidly purifying genetic resources and for producing double haploid (DH) homozygous lines in a short period. IMC has been extensively explored in Cruciferae, especially *Brassica*, because of its straightforward operation, quick culture cycle, and vast application sectors [9]. Since Lichter (1982) successfully obtained regenerated *Brassica napus* plants using the IMC technique for the first time [10], considerable advances have been made for the successful exertion of IMC in a variety of *Brassica* crops, including cabbage (*Brassica oleracea* L. var. *capitata* L.) [11], cauliflower (*Brassica oleracea* L. var. *botrytis* L.) [12], Pakchoi (*Brassica rapa* var. *multiceps*) [13], Chinese kale (*Brassica oleracea* L. var. *alboglabra*) [14], and Chinese cabbage (*Brassica campestris* ssp. *pekinensis*) [15]. The first research on IMC-producing regenerated ornamental kale plants was achieved by Zhang et al. [16], followed by an increasing number of studies [17–19]. Nonetheless, current research on IMC of kale has mainly focused on ornamental kale varieties and the application was still challenged by the low embryo induction rate.

A series of internal and external factors has proven to constitute the essential conditions for affecting microspore embryogenesis [20–23], including donor-plant genotype, media constituents, developmental stages of donor microspore, stress treatments, and culture conditions. A key internal factor is the genotype of donor material. Under the same external conditions, plants with different genotypes in the same species may show varying abilities in microspore embryogenesis [24–27]. After decades of exploratory optimization of multi-species IMC systems, another consensus is that appropriate application of plant growth regulators (PGRs) in culture medium can effectively increase microspore embryogenesis by promoting cell growth and division, mainly containing cytokinin (6-benzylaminopurine (6-BA), thidiazuron (TDZ), kinetin (KT), zeatin (ZT), etc.) and auxin (indole-3-acetic acid (IAA), 1-naphthaleneacetic acid (NAA), 2,4-dichlorophenoxyacetic acid (2,4-D), 3-indole butyric acid (IBA), etc.) [28–30]. For example, Ahmadi et al. [31] investigated the effective induction of microspore embryogenesis in the *Brassica napus* using abscisic acid (ABA). Compared to the control group, ABA (0.5 mg/L for 12 h) increased normal plantlet regeneration by 68% and increased microspore embryogenesis by nearly three times. Gu et al. [32] found that 4.6 µM ZT and 0.12 µM IAA could prompt *Brassica nigra* plantlet regeneration, and the doubling rate can reach 50.6%. On top of these, the cytokinin-like substance TDZ has recently emerged as a highly efficient bioregulator utilized in plant tissue culture. It can significantly promote cell proliferation at low concentrations, but it has been studied relatively little in microspore culture [33–35]. Studies known to date have successfully explored appropriate doses of TDZ in a few species, resulting in better somatic or microspore embryogenesis rates [36–38]. Although PGRs have successfully been explored in the microspore culture of several *Brassica* plants, there are few reports regarding kale [39,40]. To this end, the application effect of plant growth regulators in the microspore embryogenesis of kale deserves further investigation.

Given that the application of high-efficiency IMC in kale is still restricted by the low embryogenesis rate, the present study was, therefore, undertaken to evaluate the effects of different concentrations of the cytokinin TDZ on microspore embryogenesis induction and plant regeneration in two edible curly kale cultivars, for the first time. The ploidy of regenerated plantlets was detected by flow cytometry (FCM) and the abundant horticultural

characteristics of the obtained DH lines were evaluated. This comprehensive study aims to improve the IMC system for kale and contribute to the practical application of IMC technology in breeding improvement and variety purification.

## 2. Materials and Methods

### 2.1. Plant Materials

Two commercial varieties of edible curly kale, the 'Starbor $F_1$' and 'Winterbor $F_1$', were purchased from Bejo zaden, B.V. (https://www.bejoseeds.com/kale, accessed on 1 May 2019), whereafter their seeds were sown in 50-hole plug trays on 15 July 2019. The 30-day-old seedlings were then colonized in the experimental pots in the plastic greenhouse of the Genetics and Breeding of Cruciferous Vegetable Crops Laboratory Base of Shenyang Agricultural University, Liaoning Province, China. The plants were transferred into greenhouse in mid to late November. In March of the following year, on a sunny morning, after three or four inflorescence branches covered with pollination bags had a sufficient number of flowers open, the buds to be released were gently broken with tweezers to expose the stigmas, and pollen from the open flowers was applied to the stigmas to harvest $F_2$ generation seeds. On 16 August 2020, the $F_2$ seedlings were also planted in 50-hole plug trays before being colonized in experimental field. From the $F_2$ population of 'Starbor $F_1$' and 'Winterbor $F_1$' (Figure 1), the $F_2$ plants with phenotype characteristics as similar as possible to their respective $F_1$ commercial variety were selected as donor plants for IMC. Specifically, the 'Starbor $F_2$' and the 'Winterbor $F_2$' plants exhibiting vigorous growth with compact multiple branches, dark green leaves, curled leaf margins, and free from disease and wilting were chosen. The overall height of 'Winterbor $F_2$' plants was relatively slightly higher than the 'Starbor $F_2$' plants. The plants were grown in the greenhouse after being put into plastic flower pots. After the appearance of flower buds, regular fertilizer and water management are required. The upper inflorescence is clipped after bolting to encourage the development of lateral shoots.

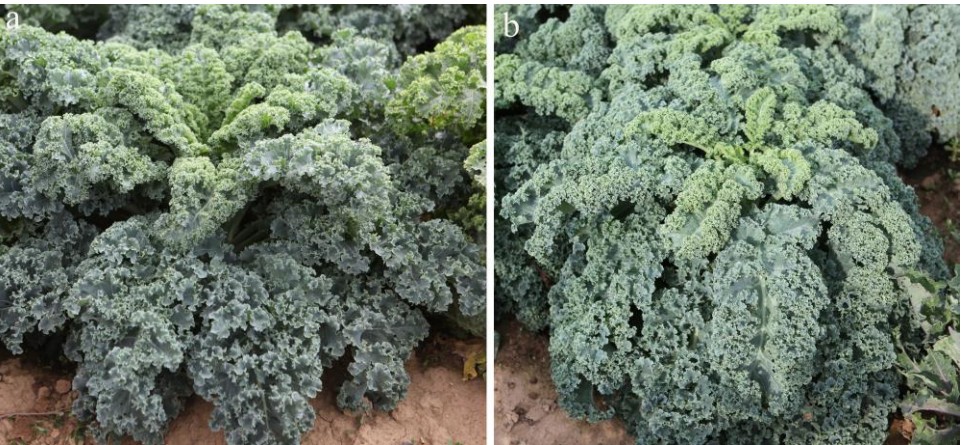

**Figure 1.** Two commercial varieties of edible kale used in this experiment self-pollinated for subsequent IMC-generating $F_2$ donor plants: (**a**) 'Starbor $F_1$'. (**b**) 'Winterbor $F_1$'.

### 2.2. Isolation of Microspores

The procedure of microspore culture was conducted as the modified version by Sato et al. [41] and Zhang et al. [16], and the buds from test material plants were taken on a sunny forenoon. We collected flower buds with the ratio of petal length to anther length being 0.50–0.75 from robust branches, and each inflorescence had at least 6 flower buds. The buds were washed with 75% ethanol for 30 s, 0.1% chlorinated mercury for 6 min, and then three times for 5 min each with sterile distilled water. After completing the above steps, buds were mashed in 10–15 mL $B_5$ medium (pH = 5.48, with 13% ($w/v$) sucrose). Using a 40 μm cell strainer and a sterile glass rod, the microspores were separated from buds, and then the filtered microspores were collected in 50 mL centrifuge tube by

centrifugation at 2000 rpm for 3 min, from which the supernatant was discarded and the precipitate was rinsed twice in fresh $B_5$ culture medium. Then, we diluted the microspores with the NLN liquid culture medium supplemented with 13% (*w/v*) sucrose (NLN-13 culture medium, pH = 5.48) so that the final cell density reached $1 \times 10^5$–$2 \times 10^5$ under a hemocytometer, and then added 1/50 volume of activated carbon in NLN-13 culture medium (liquid activated carbon is a mixture of 0.5 g/L agarose and 10 g·L$^{-1}$ activated carbon, autoclaved). The microspore suspension was then transferred in 5 mL into sterile plastic culture dishes. The Petri plates were incubated via a 24 h heat shock treatment at 33 °C before being grown for 20–23 days at 25 °C in a dark box. Once the embryos grew to about 0.3–0.5 cm and could be visible (approximately 21 days after IMC), we moved the Petri plates into a slow rotating shaker at 25 °C and 50 rpm. The embryos progressively differentiated in continuous darkness until the cotyledon shape was completely formed.

### 2.3. Effect of Genotypes on the Embryogenesis

The embryonic number derived from 'Starbor $F_2$' and 'Winterbor $F_2$' was recorded after cultivation for 30 days, and the difference in microspore embryogenesis between the genotypes was analyzed.

### 2.4. TDZ Treatments

Thidiazuron was dissolved in dimethyl sulfoxide (DMSO) to reach a concentration of 0.5 mM, and further adjusted to pH 5.80 serving as the stock solution. The tested concentration for TDZ solution is 0 mg/L, 0.1 mg/L, 0.2 mg/L, 0.3 mg/L, 0.4 mg/L, and 0.5 mg/L. The 0 mg/L TDZ was used as the control. Corresponding volumes of TDZ were added to NLN-13 medium to set different concentration gradients, which together with the microspore suspensions were subpackaged into Petri dishes. The test was repeated three times for each concentration gradient, 30 Petri dishes at a time.

### 2.5. Embryo Germination and Plantlet Regeneration

The microspore embryogenesis rate of two genotypes of edible kale was detected when the embryos appeared and reached cotyledon stage, and the number of embryos was recorded. Afterward, all strong cotyledon shaped embryos were transplanted to solid MS medium, which was subject to autoclaved sterilization with a pH of 5.80 to 5.84, containing 13% sucrose, 0.55% agar, and 0.1% activated carbon and cultivated at 25 °C with a photoperiod of 16/8 h. After the callus was transferred to MS medium, subculture was conducted every month until the growing point developed, and subculture was conducted again before the strong seedlings took root. The rates of direct transformation of embryos into plants, callus formation, and mortality rate were recorded.

### 2.6. Effect of NAA on Root Regeneration

The young regenerated rootless seedlings were transferred to MS and 1/2 MS solid medium, and various concentrations of NAA (0.1 mg/L, 0.2 mg/L, 0.3 mg/L) were added 25 days before transplantation. The common formulation whereby 0.1 mg/L NAA was added to MS medium served as the control [42,43]. Ten plants were treated at each concentration and repeated three times. The rooting of seedlings was monitored and counted after 20 to 25 days.

### 2.7. Ploidy Identification of Regenerated Plants

The ploidy of the regenerated microspore plants was identified by flow cytometry (BD Biosciences, Franklin Lakes, NJ, USA). First, samples of the newest growing leaves were taken from the field and crushed and frozen using liquid nitrogen. After the last grinding, add 1 to 2 mL of chopping buffer, then let stand for 8 min as previously described [19]. Filter through 300-mesh sieve to a 1.5-centrifugal tube and centrifuge (1000 rpm, 10 min). The supernatant was removed and treated with 500 mL of propidium iodide (PI) for 15 min in the dark. A 500-mesh screen was used to filter the mixtures into a new sample

tube, whereafter flow cytometry was used to identify DNA ploidy. The DNA of standard diploid plants was measured as the control, and the ploidy was determined according to the isolated peaks with different fluorescence intensities, and these were divided into diploid, haploid, and tetraploid (100, 200, and 400 on the *X*-axis, respectively).

### 2.8. Horticultural Characteristics of the DH Lines

As the fresh and tender green leaves are the perfect edible area, the plant growth directly affects its yield. Thus, after two months of culture in a nutrient bowl under optimal soil moisture conditions, the DH lines regenerated from two edible kale varieties 'Winterbor $F_2$' and 'Starbor $F_2$' were transplanted to the field to identify their horticultural characteristics. A completely randomized design was adopted and repeated three times. The plant height, plant width, maximum leaf length, maximum leaf width, leaf shape, leaf color, petiole length, and petiole width of DH lines regenerated plants were measured.

### 2.9. Data Collection and Statistical Analysis

In the IMC experiment, each TDZ treatment with a different concentration was repeated in triplicate. The number of embryos was counted from the 21st day of microspore culture. When the embryos appeared and reached cotyledon-shaped stage, the microspore embryogenesis ability of two genotypes was detected, and the number of embryos in each bud was recorded. We calculated the number of embryos in each bud, the proportion of regenerated embryos that directly formed the plant, and the proportion of calli that indirectly formed the plant. All the experimental data were analyzed using analysis of variance (ANOVA), and Duncan Multiple Range Test (DMRT) ($p \leq 0.05$) in SPSS 26.0 software was used for statistical evaluation to estimate the significance of differences among treatments.

## 3. Results

### 3.1. Donor Genotype Is the Primary Intrinsic Factor for the Microspore Embryogenesis

When using NLN-13 media without TDZ addition for the IMC of 'Starbor $F_2$' and 'Winterbor $F_2$' as the control, both genotypes of the curly kale successfully generated microspore-derived embryos (Table 1). The induction rate of 'Starbor $F_2$' (8.00 embryos per bud) was about three times higher than that of 'Winterbor $F_2$', which had a low induction ratio of 2.67 embryos per bud. Notably, the application of TDZ can effectively promote microspore embryogenesis within the range of appropriate concentrations of these two genotypes, but their embryogenesis rates were obviously different. These results demonstrated that the donor genotype is the key internal determinant of microspore embryogenesis.

**Table 1.** Effect of TDZ concentration on microspore embryogenesis in two genotypes of curly kale.

| Genotype | Concentration of TDZ (mg/L) | No. of Embryos Per Bud |
|---|---|---|
|  | 0 | 8.00 ± 0.56 c |
|  | 0.1 | 12.33 ± 0.33 ab |
| Starbor $F_2$ | 0.2 | 14.67 ± 1.45 a |
|  | 0.3 | 10.33 ± 0.74 bc |
|  | 0.4 | 10.00 ± 0.00 bc |
|  | 0 | 2.67 ± 0.33 b |
|  | 0.1 | 5.33 ± 0.33 a |
| Winterbor $F_2$ | 0.2 | 4.33 ± 0.88 ab |
|  | 0.3 | 3.00 ± 0.58 b |
|  | 0.4 | 2.71 ± 0.64 b |

Different letters in the same column denote significant differences ($p \leq 0.05$).

### 3.2. Effects of TDZ on Microspore-Derived Embryogenesis

To estimate the effect of TDZ on microspore embryogenesis, the control together with four incremental concentrations of TDZ was applied to the two genotypes of curly kale,

respectively. As depicted in Table 1 and Figure 2, compared with the control media, the application of TDZ can promote the embryo induction rate of two genotypes of curly kale to varying degrees. When employing 0.2 mg/L TDZ to 'Starbor $F_2$', the embryogenesis frequency reached the highest, which was 14.67 embryos per bud, 1.83-fold higher than that in the control group (Table 1, Figure 2c). The maximum induction rate of 'Winterbor $F_2$' embryos was 5.33 embryos per bud, doubled compared to the control group (Table 1, Figure 2g), which was achieved at a TDZ dose of 0.1 mg/L followed by 0.2 mg/L (1.62-fold increase). By contrast, TDZ treatments at concentrations of 0.3 and 0.4 mg/L resulted in slightly increased microspore-derived embryos compared to the control. These results indicated that proper TDZ treatment could significantly promote the formation of curly kale microspore embryos (Table 1).

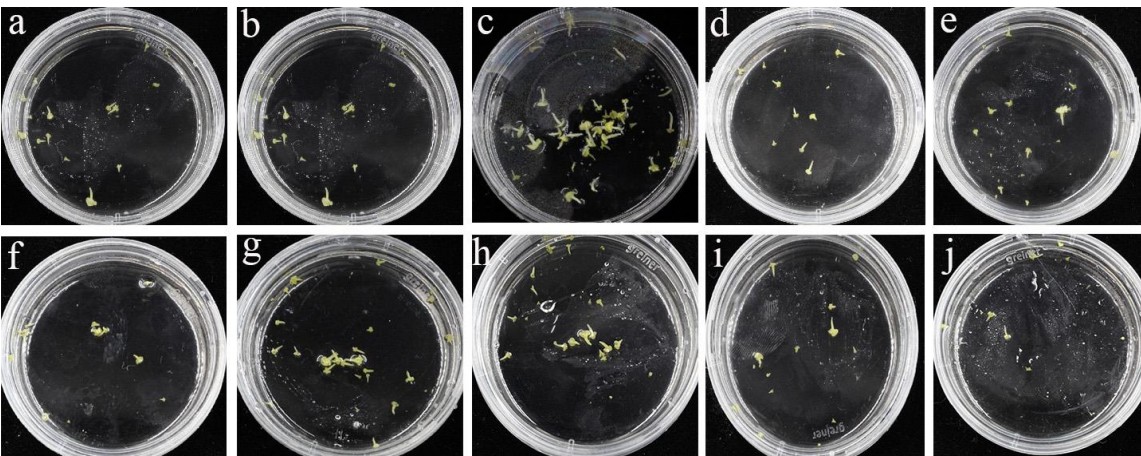

**Figure 2.** The microspore-derived embryo yield corresponded to incremental TDZ concentrations of two genotypes of curly kale: (**a**) The embryos of genotype 'Starbor $F_2$' yielded without TDZ addition. (**b**–**e**) The embryos of genotype 'Starbor $F_2$' yielded in the presence of TDZ. (**f**) The embryos of genotype 'Winterbor $F_2$' yielded without TDZ addition. (**g**–**j**) The embryos of genotype 'Winterbor $F_2$' yielded in the presence of TDZ. The TDZ concentrations for each row from left to right were 0, 0.1, 0.2, 0.3, and 0.4 mg/L.

*3.3. An Optimal Effect on Plantlet Regeneration was Achieved with 0.2 mg/L TDZ*

In the subsequent embryo developmental stage, every six embryos were implanted in MS solid medium (Figure 3), from which a part of the embryos would grow into an irregular mass of calli that usually takes 2–3 generations of continuous succession to make seedlings (Figure 3b,d), while the cotyledon-shaped embryos could directly develop into re-generated plants (Figure 3a,c). Even though the two genotype-different curly kale cultivars, 'Winterbor $F_2$' and 'Starbor $F_2$', displayed distinct plant regeneration capacities (Table 2, Figure 3), there was the similarity that the embryoid bodies of 'Winterbor $F_2$' and 'Starbor $F_2$' both exhibited high callus-formed rates with difficult plant regenerations. Compared to the initial significantly different embryogenesis rate that resulted from genotype differences, we observed that the trends of embryo survival, direct plant conversion, and callus conversion were highly similar between the two test materials under the same TDZ treatments. Furthermore, when treated with 0.2 mg/L TDZ, the 'Starbor $F_2$' embryos showed the lowest death rate of 17.6%, and the highest rates of callus formation and direct plant regeneration, which were 55.29% and 27.10%, respectively (Table 2). For genotype 'Winterbor $F_2$', the proportion of embryos directly developing into seedlings peaked at 33.13% when the TDZ concentration was 0.2 mg/L, which was also accompanied by almost the highest callus formation rate of 53.08% and the lowest death rate of 13.17%. Taken together, 0.2 mg/L TDZ significantly enhanced plantlet regeneration as the optimal concentration for both genotypes of curly kale.

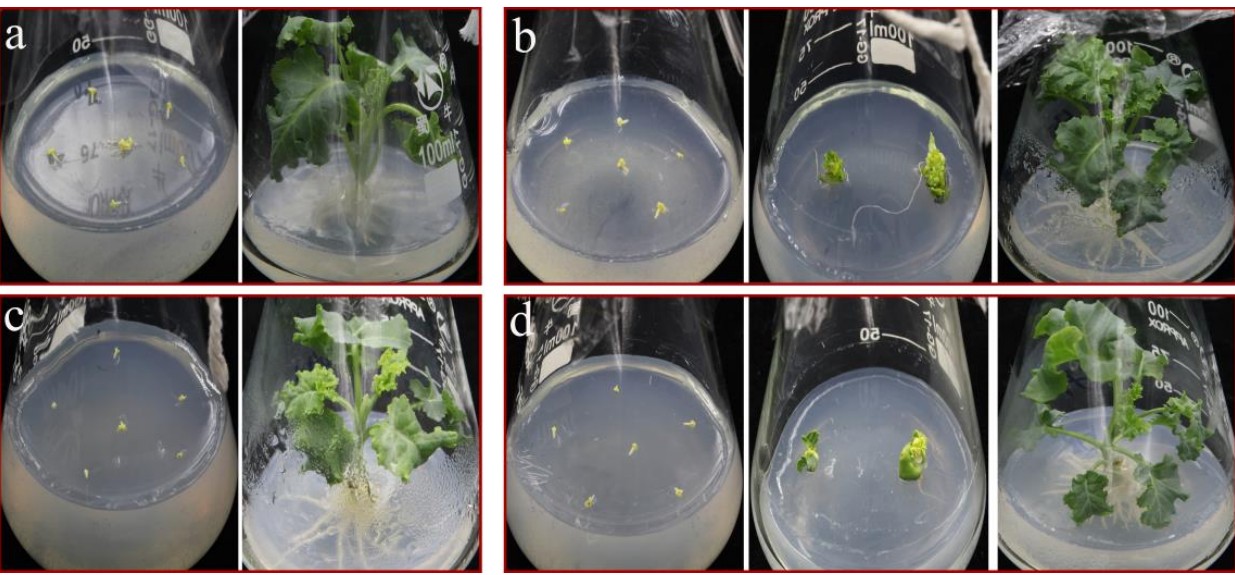

**Figure 3.** Two types of developmental processes from embryos to regenerated plantlets of 'Starbor F$_2$' and 'Winterbor F$_2$' on MS solid medium: (**a**) The cotyledon shaped embryos of 'Starbor F$_2$' directly developed into regenerated plants. (**b**) The cotyledon shaped embryos of 'Starbor F$_2$' developed into irregular calli and then formed regenerated plants. (**c**) The cotyledon shaped embryos of 'Winterbor F$_2$' directly developed into regenerated plants. (**d**) The cotyledon shaped embryos of 'Winterbor F$_2$' developed into irregular calli and then formed regenerated plants.

**Table 2.** Effect of TDZ on plant regeneration for curly kale.

| Genotype | Concentration (mg/L) | Rate of Directly Conversion to Seedlings (%) | Rate of Embryos Conversion to Callus (%) | Rate of Embryos Death (%) |
|---|---|---|---|---|
| Starbor F$_2$ | 0 | 16.83 ± 0.08 b | 36.46 ± 1.23 c | 46.73 ± 0.35 bc |
| | 0.1 | 14.36 ± 0.08 c | 15.74 ± 1.31 e | 69.89 ± 0.60 a |
| | 0.2 | 27.10 ± 0.29 a | 55.29 ± 0.16 a | 17.6 ± 0.38 d |
| | 0.3 | 14.13 ± 2.88 c | 39.50 ± 0.96 b | 46.42 ± 0.71 c |
| | 0.4 | 9.85 ± 0.22 d | 32.97 ± 0.36 d | 57.2 ± 0.48 b |
| Winterbor F$_2$ | 0 | 15.77 ± 2.87 bc | 33.13 ± 2.58 b | 51.11 ± 1.11 a |
| | 0.1 | 21.53 ± 2.06 b | 31.27 ± 0.49 b | 47.10 ± 1.92 a |
| | 0.2 | 33.13 ± 1.31 a | 53.08 ± 0.43 a | 13.17 ± 0.74 c |
| | 0.3 | 21.48 ± 0.74 b | 54.82 ± 0.74 a | 23.67 ± 1.67 b |
| | 0.4 | 11.70 ± 1.76 c | 35.15 ± 1.21 b | 52.78 ± 2.78 a |

Different letters in the same column denote significant differences ($p \leq 0.05$).

*3.4. Half Strength MS Medium Is Beneficial to the Rooting of 'Starbor F$_2$' Regenerated Seedlings*

In the rooting stage, it was observed that the regenerated plants of two genotypes of curly kale were difficult to take root. To address this issue, we selected 'Starbor F$_2$' seedlings, most of which failed to take root, as the test material for the rooting rate. Compared with the control (0.1 mg/L NAA + MS), by adding 0.2 and 0.3 mg/L NAA to MS medium, we found that there was no significant difference in the rooting rate among the treated seedlings (Table 3). By contrast, in 1/2 MS media, 0.1 to 0.2 mg/L NAA was beneficial to root development. As shown in Figure 4b, abundant roots were produced in 1/2 MS media and distributed evenly, among which the rooting rate reached 93.33% at the concentration of 0.2 mg/L NAA, which was significantly higher than that at other concentrations (Table 3, Figure 4a). These results demonstrated that the 1/2 MS medium had a superior rooting effect for curly kale.

**Table 3.** Effects of MS and 1/2 MS medium with different concentrations of NAA on the rooting of regenerated plantlets.

| NAA (mg/L) | Number of Observed Plants | Rooting Rate | |
|---|---|---|---|
| | | **MS** | **1/2 MS** |
| 0.1 | 90 | 60.00 ± 0.00 a | 83.33 ± 0.33 ab |
| 0.2 | 90 | 63.33 ± 0.12 a | 93.33 ± 0.33 b |
| 0.3 | 90 | 71.67 ± 0.83 a | 76.67 ± 0.67 a |

Different letters represent significant differences ($p \leq 0.05$).

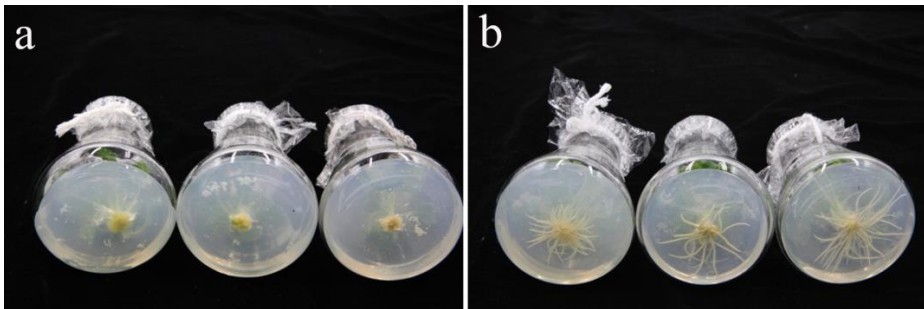

**Figure 4.** The effects of MS and 1/2 MS medium on the rooting of regenerated plants under the treatment of 0.2 mg/L NAA: (**a**) Normal regenerated seedlings were difficult to take root in 100% MS medium. (**b**) The 1/2 MS medium containing 0.2 mg/L NAA could effectively promote the rooting of normal seedlings.

*3.5. The Tested Genotypes Exhibited Similar Chromosome Doubling Efficiencies*

The regenerated plantlet population of edible kale derived from IMC was comprised of mixed plant ploidy. Flow cytometry (FCM) was employed to detect the ploidy level of regenerated microspore plantlets from two genotypes of curly kale. Using the common diploid as a control, its peak value corresponded to the vicinity of the abscissa 200. The peak value achieved at abscissas 100, 200, and 400 from Figure 5a–c that referred to the regenerated plant was a haploid, a diploid, and a polyploid, respectively. For 'Starbor $F_2$' and 'Winterbor $F_2$', a total of 663 and 253 regenerated plants for each were obtained, from which 61 randomly selected regenerated plants were used to detect the ploidy, separately. The double diploid rate of 'Winterbor $F_2$' and 'Starbor $F_2$' was 39.34% and 32.78%, and the polyploids accounted for the least, with 3.78% and 4.92%, respectively (Table 4). For both, the proportion of haploids was the highest, and the haploid rate of 'Starbor $F_2$' was relatively higher than that of 'Winterbor $F_2$'. Overall, there was no obvious difference in doubling efficiency between the two genotypes of edible curly kale.

*3.6. Horticultural Characteristics of the DH Lines*

To evaluate the horticultural characteristics of obtained DH lines germplasm resources, the plant height, plant width, leaf length, leaf shape, leaf color, and petiole of DH lines from 'Winterbor $F_2$' and 'Starbor $F_2$' were measured (Table 5, Figures 6 and 7). The results showed that the average plant height for 'Starbor $F_2$' was 21.11 cm, with a variable range of 16.00–26.00 cm. The average plant width was 34.43 cm, with a fluctuation range of 33.00–37.00 cm. The range of leaf lengths was 9.50 to 14.00 cm, with 11.97 cm being the average. The average leaf width was 8.44 cm, with a range of 7.50 to 9.20 cm. The average petiole length was 8.33 cm, with a range of 7.50 to 9.50 cm. The leaf shape was divided into highly curled leaf and moderately curled margin, and the leaf color was divided into dark green and green. The average plant height for 'Winterbor $F_2$' was 21.89 cm, with a fluctuation range of 15.00–30.00 cm. The average plant width was 37.61 cm, with a fluctuation range of 33.00–42.00 cm. The average leaf length was 13.06 cm, with maximum leaf lengths between 10.50 and 15.00 cm. The average leaf width was 6.83 cm, while the maximum leaf width ranges from 6.50 to 7.50 cm. The average petiole length was 11.61 cm, while the range was 10.50–13.00 cm. Slender curly leaves and

curly leaves with a circular edge made up the leaf shape, while green and yellow-green distinguished the leaf color. Taken together, there were abundant variations in horticultural characteristics of DH strains of 'Winterbor $F_2$' and 'Starbor $F_2$'.

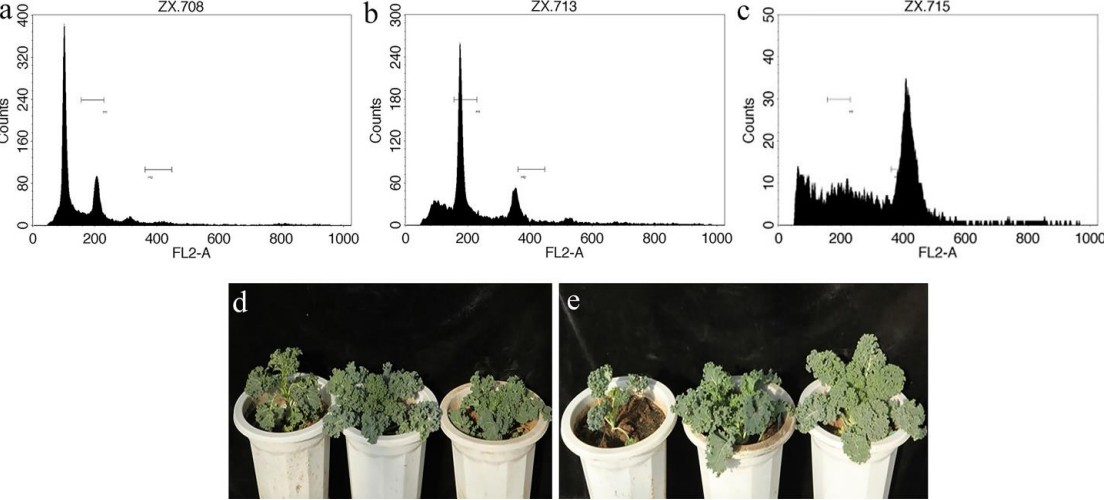

**Figure 5.** Ploidy identification by flow cytometry and the regenerated plants of haploid doubled haploids, and polyploids: (**a**) DNA content distribution of haploid plants. (**b**) DNA content distribution of doubled haploid plants. (**c**) DNA content distribution of polyploid plants. (**d**) Haploid, double haploid, and polyploid regenerated plants of 'Starbor $F_2$' in sequence from left to right. (**e**) Haploid, double haploid, and polyploid regenerated plants of 'Winterbor $F_2$' in sequence from left to right.

**Table 4.** Ploidy identification of regenerated plants from the two genotypes of curly kale.

| Genotype | Number of Observed Plants | Number of Haploid Plants | Number of Double Haploid Plants | Number of Polyploid Plants | Doubling Efficiency (%) |
|---|---|---|---|---|---|
| Starbor $F_2$ | 61 | 41 | 18 | 2 | 32.78 |
| Winterbor $F_2$ | 61 | 34 | 24 | 3 | 39.34 |

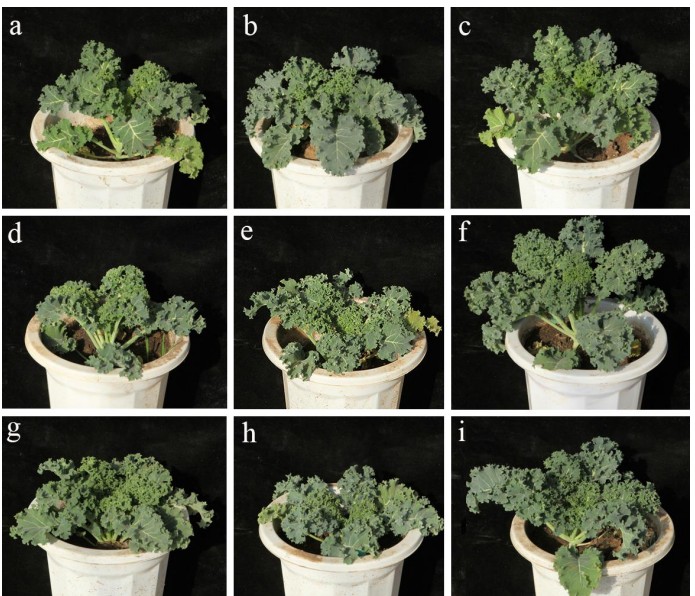

**Figure 6.** The DH lines regenerated from 'Starbor $F_2$': (**a**) 'S1'. (**b**) 'S2'. (**c**) 'S3'. (**d**) 'S4'. (**e**) 'S5'. (**f**) 'S6'. (**g**) 'S7'. (**h**) 'S8'. (**i**) 'S9'.

**Table 5.** Horticultural characteristics of doubled haploid lines of edible curly kale.

| DH Lines | Variety Source | Plant | | The Maximum Leaf | | Petiole | | Leaf | |
|---|---|---|---|---|---|---|---|---|---|
| | | Height (cm) | Width (cm) | Length (cm) | Width (cm) | Length (cm) | Width (cm) | Shape | Color |
| S1 | Starbor $F_2$ | 24.00 | 33.00 | 13.50 | 9.20 | 8.50 | 0.70 | highly curled | green |
| S2 | Starbor $F_2$ | 24.00 | 34.00 | 12.70 | 7.50 | 8.00 | 0.80 | highly curled | dark green |
| S3 | Starbor $F_2$ | 23.50 | 37.00 | 12.00 | 8.00 | 9.00 | 0.70 | highly curled | green |
| S4 | Starbor $F_2$ | 22.00 | 33.00 | 10.50 | 7.80 | 9.50 | 0.50 | highly curled | green |
| S5 | Starbor $F_2$ | 17.00 | 33.50 | 10.00 | 8.50 | 8.50 | 0.60 | highly curled | dark green |
| S6 | Starbor $F_2$ | 26.00 | 37.00 | 14.00 | 9.00 | 7.50 | 0.60 | highly curled | dark green |
| S7 | Starbor $F_2$ | 17.50 | 33.00 | 11.50 | 8.50 | 8.00 | 0.70 | Moderately curled margin | dark green |
| S8 | Starbor $F_2$ | 16.00 | 32.40 | 9.50 | 8.50 | 8.00 | 0.50 | Moderately curled margin | dark green |
| S9 | Starbor $F_2$ | 20.00 | 37.00 | 14.00 | 9.00 | 8.00 | 0.70 | highly curled | dark green |
| | Mean | 21.11 | 34.43 | 11.97 | 8.44 | 8.33 | 0.64 | | |
| | Min | 16.00 | 32.40 | 9.50 | 7.50 | 7.50 | 0.50 | | |
| | Max | 26.00 | 37.00 | 14.00 | 9.20 | 9.50 | 0.80 | | |
| | SD | 3.61 | 1.97 | 1.71 | 0.58 | 0.61 | 0.10 | | |
| D1 | Winterbor $F_2$ | 17.00 | 33.00 | 10.50 | 7.50 | 11.00 | 0.60 | Circular edge curly | green |
| D2 | Winterbor $F_2$ | 26.00 | 37.00 | 14.00 | 6.50 | 11.00 | 0.60 | Slender curly | green |
| D3 | Winterbor $F_2$ | 26.00 | 35.00 | 11.50 | 6.50 | 11.00 | 0.70 | Circular edge curly | green |
| D4 | Winterbor $F_2$ | 21.00 | 40.50 | 13.50 | 7.50 | 12.00 | 0.80 | Slender curly | yellow green |
| D5 | Winterbor $F_2$ | 16.00 | 37.00 | 13.50 | 6.50 | 11.00 | 0.70 | Circular edge curly | yellow green |
| D6 | Winterbor $F_2$ | 30.00 | 42.00 | 15.00 | 6.50 | 13.00 | 0.60 | Slender curly | yellow green |
| D7 | Winterbor $F_2$ | 26.00 | 40.00 | 14.00 | 6.50 | 12.00 | 0.60 | Slender curly | yellow green |
| D8 | Winterbor $F_2$ | 20.00 | 36.00 | 11.00 | 7.50 | 13.00 | 0.70 | Circular edge curly | green |
| D9 | Winterbor $F_2$ | 15.00 | 38.00 | 14.50 | 6.50 | 10.50 | 0.60 | Slender curly | yellow green |
| | Mean | 21.89 | 37.61 | 13.06 | 6.83 | 11.61 | 0.66 | | |
| | Min | 15.00 | 33.00 | 10.50 | 6.50 | 10.50 | 0.60 | | |
| | Max | 30.00 | 42.00 | 15.00 | 7.50 | 13.00 | 0.80 | | |
| | SD | 5.33 | 2.85 | 1.63 | 0.50 | 0.93 | 0.07 | | |

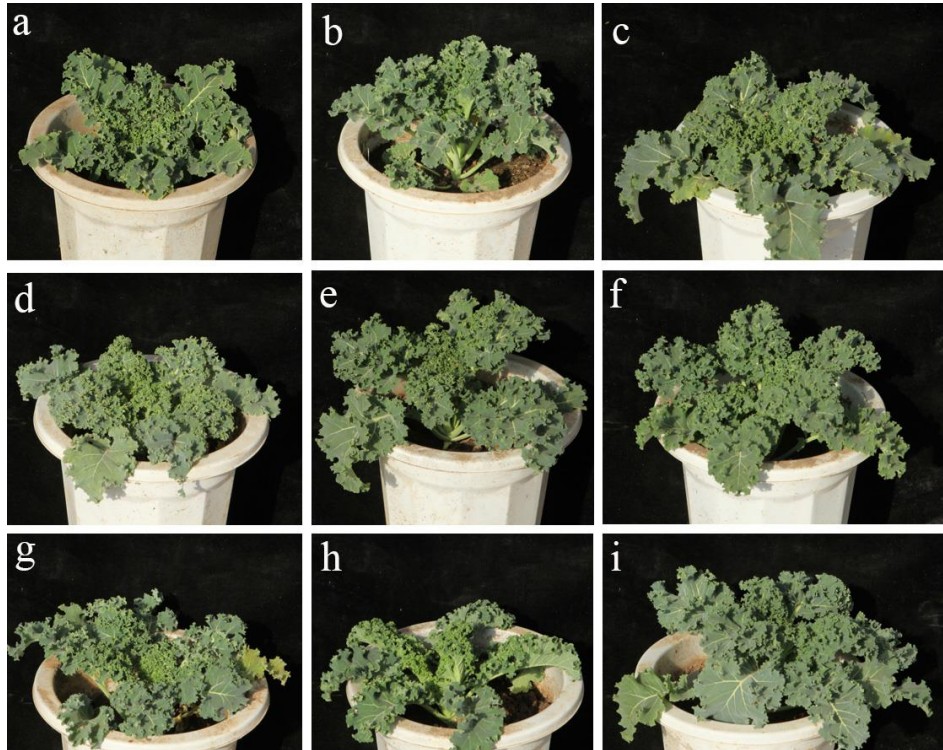

**Figure 7.** The DH lines regenerated from 'Winterbor $F_2$': (**a**) 'D1'. (**b**) 'D2'. (**c**) 'D3'. (**d**) 'D4'. (**e**) 'D5'. (**f**) 'D6'. (**g**) 'D7'. (**h**) 'D8'. (**i**) 'D9'.

## 4. Discussion

　　IMC is an effective technique to obtain homozygous DH lines in haploid breeding, which can greatly improve purification efficiency and shorten the plant breeding process. It has made great progress and has been widely applied in *Brassica* crops over the last decade, as well as in eggplant [44], wheat [45], radish [46], and alfalfa [47]. Continuous exploration and improvement in the procedure of IMC have achieved remarkable results, especially in the regeneration of recalcitrant-genotype plants [20,28,48]. For kale, current reported studies mainly focus on establishing more efficient protocols of embryogenesis and chromosome doubling. In the present study, we succeeded in producing embryos and DH plants through TDZ-treated microspore culture in two genotypes of curly kale, which provided a basis for the practical application of microspore culture in future breeding programs.

　　Previous studies have shown that the trigger mechanism of microspores embryogenesis is highly complex and easily affected by several factors, among which the most fundamental is donor-plant genotypes that will lead to significant discrepancies in the embryogenesis frequency [24,49–51]. Therefore, in our experiment, we first selected two genotypes of edible kale as test materials and performed the same treatments, first examining the effect of genotypic differences. The results showed that there was a nearly twofold difference in the embryogenesis rates between the 'Starbor $F_2$' (8.00 embryos per bud) and 'Winterbor $F_2$' (2.67 embryos per bud) in their respective control treatments (Table 1). Thus, the induction frequency of microspore embryos was closely correlated with the genotypes in edible kale, which was consistent with previous studies in *B. oleracea*. Zhao et al. [27] explored the embryo emergence rate of 13 cultivars of broccoli. Though all the cultivars were successfully induced, there was a significant difference among 13 genotypes, with the highest value reaching 3.15 and the lowest value being 0.28. Likewise, Zhang et al. [16] studied the embryo emergence rate of ornamental kale using 29 different genotypes as donor plants. Only six genotypes induced embryos, which corresponded to the more out-bred genotypes. Furthermore, in previous studies to explore the effect of chemical treatment (suberoylanilide hydroxamic acid, L-ascorbic acid sodium salt, and trichostatin A) on microspore embryogenesis in ornamental kale, the cultivars that produced more microspore-derived embryos in the control group still showed a relatively higher embryogenesis rate than other low-response genotypes under the same chemical treatment [17,19,40]. In agreement with previous results, a higher number of microspore-derived embryos was observed in 'Starbor $F_2$' than in 'Winterbor $F_2$' across all concentrations of the TDZ treatment, suggesting that high-response 'Starbor $F_2$' may be more advantageous for practical applications. Moreover, several studies also discovered that plant purity had an impact on embryogenesis, namely that the $F_2$ inbred line had the highest embryo emergence rate, while the $F_5$ inbred line had the lowest [52]. Consequently, it was considered that the purer the germplasm resources of kale, the lower the frequency of microspore embryos. For this purpose, we selected phenotypically superior plants from the $F_2$ generations of the two genotypes of curly kale for IMC. It could allow us to obtain a relatively high microspore embryogenesis rate and cultivate abundant DH lines from them to meet our breeding objectives. Taken together, we believed that the genotype has a fundamental effect on the embryo occurrence rate of edible kale, and it is recommended to first test the response of different genotypes to reasonably detect other factors.

　　According to the previous studies, adding plant growth regulators in appropriate concentrations to NLN medium is an effective means to improve microspore embryogenesis [31,53]. Adjusting the ratio of auxin to cytokinin can effectively improve the embryo induction rate. Herein, we applied a cytokinin-like substance TDZ to change the ratio of auxin to cytokinin, and further investigated its effect on the IMC of edible curly kale. It has been reported that TDZ can effectively induce somatic embryogenesis, promote the accumulation of zeatin, dihydrozeatin, adenine, and adenosine, inhibit the activity of cytokinin oxidase, lead to the accumulation of cytokinin and induce callus to sprout [29,54–56]. Cappelletti et al. [33] found that the combination of 0.02 mg/L 2,4-D and 0.5 mg/L TDZ had the highest regeneration efficiency for strawberry leaves. Another

study exploring the regeneration potential of *B. oleracea* leaf tissues showed that the optimum shoot induction (54.44%) was obtained by supplementing 4.5 µM TDZ and 0.5 µM NAA [34]. Notably, TDZ could promote callus and bud induction at low concentrations. Similarly, we also found that low-concentration TDZ could successfully improve embryo induction, while the effect of TDZ treatment was genotype-dependent (Table 1). In our experimental design, we tested a total of five TDZ concentrations (0, 0.1, 0.2, 0.3, and 0.4 mg/L), which were set up in reference to the findings of Jia et al. [37]. In their study, the optimal TDZ application concentration for microspore embryogenesis was narrowed to 0.1 mg/L to 0.5 mg/L in Chinese flowering cabbage. Based on the study, our study further determined the optimal TDZ application concentration in curly kale. In the stage of embryo rate calculation, 0.2 mg/L TDZ had the best effect on 'Starbor $F_2$', while 0.1 mg/L TDZ had the best effect on 'Winterbor $F_2$', followed by 0.2 mg/L TDZ. However, the promotion effect of TDZ on microspore embryogenesis was greatly reduced under higher concentrations (Table 1). Accordingly, we observed that at the low concentration of TDZ, the embryo mortality rate decreased with the increase in the embryogenesis rate and the direct seeding rate during the stage of plantlet regeneration (Table 2). For 'Starbor $F_2$', 0.2 mg/L TDZ treatment achieved the highest embryogenesis rate (1.83-fold higher than the control group) and direct seeding rate (1.61-fold increase), and the lowest mortality rate. Likewise, 0.2 mg/L TDZ increased the embryogenesis rate of 'Winterbor $F_2$' 1.62 times, the direct seeding rate 1.61 times, and the mortality rate fell to the lowest. Consistent with the results of Jia et al. [37], our tested TDZ concentrations had no adverse effects on embryogenesis, and 0.2 mg/L TDZ had an optimal effect on embryo survival and plant regeneration of the two genotypes. Therefore, low concentrations of TDZ added to the NLN-13 culture medium could enhance the IMC effect to obtain more DH lines, which was the first application of TDZ in the IMC of kale.

Rooting is the last critical step to ensure the survival of the regenerated plants before planting. In order to solve the problem of rooting difficulty in the regenerated plants of two genotypes of edible kale, we investigated the rooting medium to increase their rooting rate and survival rate. High-salt medium (MS) and medium containing half the inorganic salts (1/2 MS) are commonly used as the rooting medium in IMC [57]. Furthermore, auxin receptors in the formation of rhizomes have different affinities, and plant growth regulators have varying root-inducing abilities [58]. Naeem et al. [59] discovered that adding 6 mM IAA and 1 mM kinetin (KN) to MS medium developed the roots effectively. Ren et al. [60] found that the rooting rate could increase to about 95.22% when 0.2 mg/L IBA was added into 1/2 MS medium, with an average rooting number of 5.86 for *Tussilago farfara*. From a well-established quick culture system for *Lonicera macranthoides* 'Yuleil's jade Bud', it was discovered that the ideal combination for rooting medium was 1/2 MS + IBA 1.0 mg/L + NAA 0.2 mg/L [61]. Gambhir et al. [62] analyzed the in vitro-developed buds grown on MS media that contained various concentrations of IAA, NAA, and IBA. Results showed that NAA had the highest rooting rate, which was 100%, followed by IAA (97.20%) and IBA (80%). The above experimental evidence indicated that rooting effects vary from species to species. Based on our previous investigation of rooting efficiency, the addition of 0.1 mg/L NAA to MS medium has been used as a common formulation to promote rooting in *Brassica* [42,43]. In the present study, we investigated the effects of two MS mediums and different concentrations of NAA on the rooting of regenerated plants. Our findings indicated that 1/2 MS medium was more conducive to rooting, and that the combination of 1/2 MS medium + 0.2 mg/L NAA was most beneficial to robust root formation. This result was different from our commonly used rooting formulation and provided a new reference for further improving the rooting efficiency of kale.

In this study, we successfully exploited the cytokinin TDZ for improving the isolated microspore culture protocol and explored the factors influencing embryo induction and rooting for curly kale. The application of TDZ indicated that all tested TDZ concentrations had no adverse effects on embryogenesis, while low-concentration TDZ can effectively enhance the frequencies of microspore embryogenesis, and 0.2 mg/L TDZ had an optimal

effect on embryo survival and plant regeneration of the two genotypes. The average spontaneous doubling rate of the two genotypes is more than 30%, which needs to be further optimized in follow-up study to break through the bottleneck that limits the application of microspore culture technology in large-scale breeding. DH plants with rich phenotypic variations were obtained from the $F_2$ individuals of two edible kale varieties, of which combining ability and nutritive value will be evaluated by the preparation of hybrid combinations in the next step. This study provides a basis for accelerating the breeding process of new edible kale varieties, improving breeding efficiency, and better realizing market demands and breeding goals.

**Author Contributions:** J.Z. and X.Z.: writing—original draft preparation, writing—review and editing, investigation, and formal analysis. Z.G. and G.S.: software and investigation. J.R.: conceptualization, funding acquisition, and supervision. H.F.: conceptualization, project administration, supervision, and resources. All authors have read and agreed to the published version of the manuscript.

**Funding:** This research was funded by the National Natural Science Foundation of China (Grant No. 32002070).

**Data Availability Statement:** The data that support the results are included in this article. Other relevant materials are available from the corresponding author upon reasonable request.

**Conflicts of Interest:** The authors declare no conflict of interest.

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
