# Peer review of "Thidiazuron Promoted Microspore Embryogenesis and Plant Regeneration in Curly Kale (Brassica oleracea L. convar. acephala var. sabellica)"

_horticulturae, doi:10.3390/horticulturae9030327_

Round 1

Reviewer 1 Report

The manuscript “Thidiazuron promoted microspore embryogenesis and plant re- generation in curly kale (Brassica oleracea L. convar. acephala var. sabellica)” (Manuscript ID: horticulturae-2232435) shows the results of the cytokinin thidiazuron (TDZ) application in isolated microspore culture (IMC) technology with its effects on microspore embryogenesis and plant  regeneration in two curly kale cultivars (‘Winterbor F2’ and ‘Starbor F2’). The manuscript includes 6 figures and 5 tables:

Figure 1. Two commercial varieties of edible kale used in this experiment self-pollinated for subsequent IMC generating F2 donor plants. (a) ‘Starbor F1’. (b) ‘Winterbor F1’.

Figure 2. The microspore-derived embryo yield corresponded to incremental TDZ concentrations of two genotypes of curly kale. (a) The embryos of genotype ‘Starbor F2’ yielded without TDZ addition. (b-e) The embryos of genotype ‘Starbor F2’ yielded in the presence of TDZ. (f) The embryos of genotype ‘Winterbor F2’ yielded without TDZ addition. (g-j) The embryos of genotype ‘Winterbor F2’ yielded in the presence of TDZ.

Figure 3. Two types of developmental process from embryos to regenerated plantlets of ‘Starbor F2’ and ‘Winterbor F2’ on MS solid medium. (a) The cotyledon embryos of ‘Starbor F2’ directly developed into regenerated plants. (b) The cotyledon embryos of ‘Starbor F2’ developed into irregular callus and then formed regenerated plants. (c) The cotyledon embryos of ‘Winterbor F2’ directly developed into regenerated plants. (d) The cotyledon embryos of ‘Winterbor F2’ developed into irregular callus and then formed regenerated plants.

Figure 4. The effects of MS and 1/2 MS medium on the rooting of regenerated plants under the treatment of 0.2 mg/L NAA. (a) Normal regenerated seedlings were difficult to take root in 100% MS medium. (b) The 1/2 MS medium containing 0.2 mg/L NAA could effectively promote the rooting of normal seedlings.

Figure 5. Ploidy identification by flow cytometry and the regenerated plants of haploid, doubled  haploids and polyploids. (a) DNA content distribution of haploid plants. (b) DNA content distribution of doubled haploid plants. (c) DNA content distribution of polyploid plants. (d) Haploid, double haploid and polyploid regenerated plants of ‘Starbor F2’ in sequence from left to right. (e) Haploid, double haploid and polyploid regenerated plants of ‘Winterbor F2’ in sequence from left to right.

Figure 6. The DH lines regenerated from ‘Starbor F2’. (a) ‘S1’. (b) ‘S2’. (c) ‘S3’. (d) ‘S4’. (e) ‘S5’. (f)  ‘S6’. (g) ‘S7’. (h) ‘S8’. (i) ‘S9’.

Figure 7. The DH lines regenerated from ‘Winterbor F2’. (a) ‘D1’. (b) ‘D2’. (c) ‘D3’. (d) ‘D4’. (e) ‘D5’.  (f) ‘D6’. (g) ‘D7’. (h) ‘D8’. (i) ‘D9’.

Table 1. Effect of TDZ concentration on microspore embryogenesis in two genotypes of curly kale.

Table 2. Effect of TDZ on plant regeneration for curly kale.

Table 3. Effects of MS and 1/2MS medium with different concentrations of NAA on the rooting of regenerated plantlets.

Table 4. Ploidy identification of regenerated plants from the two genotypes of curly kale.

Table 5. Horticultural characteristics of doubled haploid lines of edible curly kale.

The main results are:

·         The concentrations of TDZ in regard to effect on the no. of embryos per bud in each genotype were tested, and recommendation was made.

·         The concentrations of TDZ in regard to effect on the rate of directly conversion to seedlings (%), rate of embryos conversion to callus (%), rate of embryos death (%) in each genotype, and recommendation was made.

·         Rooting rate of regenerated plantlets after application of three different concentrations of NAA  in two different mediums-MS and 1/2MS.

·         Number of haploid plants, number of double haploid plants, number of polyploid plants, doubling efficiency (%) of regenerated plants from the two genotypes of curly kale.

·         The eight horticultural traits of doubled haploid lines of edible curly kale: 1. plant height, 2. plant width, 3. maximum leaf length, 4. maximum leaf width, 5. petiole length, 6. petiole width, 7. leaf shape, 8. leaf color, were evaluated.

The manuscript’s segments to improve include:

·                Explanation how the self-compatibility was overcome during self-pollination.

·                As the effect of TDZ treatment was genotype-dependent what is strategy regarding its application? To conduct experiments for each genotype of interest? The recommendations in this research were made on the base of two genotypes.

The manuscript “Thidiazuron promoted microspore embryogenesis and plant re- generation in curly kale (Brassica oleracea L. convar. acephala var. sabellica)” (Manuscript ID: horticulturae-2232435) may be published after minor corrections.

Reviewer’s remarks

Page 1 line 12: Please insert “providing to“ after “characterized by” in abstract section.

Page 1 lines 13-14: Please delete part of sentence “which not only attracts health-concerned and fitness crowds, but” as it is not scientific language and insert “and” instead “also”.

Page 1 line 26: Please define the abbreviation “MS”.

Page 1 line 30: Please define the abbreviations “TDZ and DH lines” in the keywords.

Page 1 line 39: Please change “high nutritional densities” to more logical term.

Page 1 lines 42-43: Please delete part of the sentence “with health food and fitness crowds in recent years”.

Page 2 line 52: Please change “must be purified for” to other adequate term that will include fertilization.

Page 2 line 49: Please change “purification” to “inbreeding”.

Page 2 lines 80-81: Please give full terms for abbreviations “(6-BA, TDZ, KT, ZT, etc.) and auxin (IAA, NAA, 2,4-D, IBA)”.

Page 2 line 83: Please give the full term for “ABA”.

Page 2 line 84: Please correct “68 %“to “68%“.

Page 2 line 92: Please give the full term for PGRs after first mentioning.

Page 2 line 93: Please cite “there are few reports regarding the kale“ in regard to microspore culture.

Page 3 lines 111-113: “The plants were transferred into greenhouse in mid to late November and were self-pollinated in March of the following year to harvest F2 generation.” Please explain how the self-incompatibility was overcome.

Page 3 lines 114-117: “From the F2 population of ‘Starbor F1’ (Figure 1a), plants with leaf scroll, medium height, and robust growth were chosen, and the F2 individuals from ‘Winterbor F1’ with high growth, compact plants, and vigorous growth were chosen (Figure 1b) as donor plants for IMC.” Please explain the difference between high growth, vigorous growth and robust growth as it is unclear.

Page 3 line 130: “Mercury chloride exposure induces DNA damage, reduces fertility, and alters somatic and germline cells in Drosophila melanogaster”. (Mojica-Vázquez et al. 2019). It is questionable whether cells remained unaltered after these treatments.

Page 3 line 137: Please give the full term for the abbreviation “NLN”.

Page 3 line 125: 2.2. Isolation of microspores should be written in past tense not as lab method preparations.

Page 5 line 192: What traits were measured for petiole.

Table 1: Please add “Concentration of TDZ” in the second column.

Table 5: Please add four rows after S9 and D9: Mean, Min, Max, standard deviation or CV.

Author Response

To Reviewer 1:

Comments to the Author:

The manuscript “Thidiazuron promoted microspore embryogenesis and plant re- generation in curly kale (Brassica oleracea L. convar. acephala var. sabellica)” (Manuscript ID: horticulturae-2232435) shows the results of the cytokinin thidiazuron (TDZ) application in isolated microspore culture (IMC) technology with its effects on microspore embryogenesis and plant regeneration in two curly kale cultivars (‘Winterbor F2’ and ‘Starbor F2’). The manuscript includes 6 figures and 5 tables. The main results are:

  • The concentrations of TDZ in regard to effect on the no. of embryos per bud in each genotype were tested, and recommendation was made.
  • The concentrations of TDZ in regard to effect on the rate of directly conversion to seedlings (%), rate of embryos conversion to callus (%), rate of embryos death (%) in each genotype, and recommendation was made.
  • Rooting rate of regenerated plantlets after application of three different concentrations of NAA in two different mediums-MS and 1/2MS.
  • Number of haploid plants, number of double haploid plants, number of polyploid plants, doubling efficiency (%) of regenerated plants from the two genotypes of curly kale.
  • The eight horticultural traits of doubled haploid lines of edible curly kale: 1. plant height, 2. plant width, 3. maximum leaf length, 4. maximum leaf width, 5. petiole length, 6. petiole width, 7. leaf shape, 8. leaf color, were evaluated.

The manuscript “Thidiazuron promoted microspore embryogenesis and plant re- generation in curly kale (Brassica oleracea L. convar. acephala var. sabellica)” (Manuscript ID: horticulturae-2232435) may be published after minor corrections.

The manuscript’s segments to improve include:

  1. Explanation how the self-compatibility was overcome during self-pollination.

Response: We thank the Reviewer for bringing these to our attention. Considering the existence of self-incompatibility in Brassica during the flowering stage, we performed hand pollination for F1 plants self-crossing at the bud stage to overcome self-incompatibility and to harvest F2 seeds. The specific procedures were as follows: ​On a sunny morning, after three or four inflorescence branches covered with pollination bags had a sufficient number of flowers open, the buds to be released were gently broken with tweezers to expose the stigmas, and pollen from the open flowers was applied to the stigmas. The above related content was added to the Materials and Methods section (Lines 121-125).

  1. As the effect of TDZ treatment was genotype-dependent what is strategy regarding its application? To conduct experiments for each genotype of interest? The recommendations in this research were made on the base of two genotypes.

Response: Since the TDZ treatment was genotype-dependent, we suggested that it is a feasible strategy to compare the microspore embryogenesis rates among different genotypes in advance by the IMC with no TDZ application. In our study, we first examined the number of microspore-derived embryos for both genotypes of curly kale without TDZ application, where ‘Starbor F2’ (8.00±0.56 per bud) was significantly higher than ‘Winterbor F2’ (2.67±0.33 per bud). Then, we tested the effect of applying different concentrations of TDZ, and their embryogenesis rates increased greatly within a certain range. The higher embryogenesis rates in ‘Starbor F2’ than that in ‘Winterbor F2’ could be clearly seen. Therefore, the ‘Starbor F2’ could be more favorable for subsequent practical applications. To be clearer and in accordance with the Reviewers’ concern, we added the above relevant content to Discussion (Lines 388-396).

  1. Reviewer’s remarks:

Page 1 line 12: Please insert “providing to“ after “characterized by” in abstract section.

Response: We inserted ‘providing to’ after ‘characterized by’ in abstract section.

Page 1 lines 13-14: Please delete part of sentence “which not only attracts health-concerned and fitness crowds, but” as it is not scientific language and insert “and” instead “also”.

Response: We revised this sentence as you suggested.

Page 1 line 26: Please define the abbreviation “MS”.

Response: We added the full definition ‘Murashige and Skoog’ to the abbreviation ‘MS’.

Page 1 line 30: Please define the abbreviations “TDZ and DH lines” in the keywords.

Response: We added the ‘thidiazuron (TDZ)’ and ‘doubled haploid (DH) lines’ in the keywords.

Page 1 line 39: Please change “high nutritional densities” to more logical term.

Response: We changed the ‘high nutritional densities’ to ‘high nutritional value’ (Line 40).

Page 1 lines 42-43: Please delete part of the sentence “with health food and fitness crowds in recent years”.

Response: We deleted the ‘with health food and fitness crowds in recent years’ in this sentence (Line 44).

Page 2 line 52: Please change “must be purified for” to other adequate term that will include fertilization.

Response: We changed “must be purified for” to ‘require multiple generations of geitonogamy to obtain pure inbred lines’ (Line 57).

Page 2 line 49: Please change “purification” to “inbreeding”.

Response: We changed ‘purification’ to ‘inbreeding’ according to your suggestion (Line 62).

Page 2 lines 80-81: Please give full terms for abbreviations “(6-BA, TDZ, KT, ZT, etc.) and auxin (IAA, NAA, 2,4-D, IBA)”.

Response: We supplemented the full terms ‘6-benzylaminopurine (6-BA), thidiazuron (TDZ), kinetin (KT), zeatin (ZT)’ and ‘indole-3-acetic acid (IAA), 1-naphthaleneacetic acid (NAA), 2,4-dichlorophenoxyacetic acid (2,4-D), 3-indole butyric acid (IBA)’ for abbreviations “(6-BA, TDZ, KT, ZT, etc.) and auxin (IAA, NAA, 2,4-D, IBA)” (Lines 88-90).

Page 2 line 83: Please give the full term for “ABA”.

Response: We added the full term for ‘ABA’ (Line 92).

Page 2 line 84: Please correct “68 %“to “68%“.

Response: We corrected it to the right form ‘68%’ (Line 93).

Page 2 line 92: Please give the full term for PGRs after first mentioning.

Response: The full term for PGRs appeared at page 2 line 86 in the manuscript.

Page 2 line 93: Please cite “there are few reports regarding the kale“ in regard to microspore culture.

Response: We cited the relevant studies from Niu et al. and Liu et al. [39,40] in line 103 and revised corresponding references.

Page 3 lines 111-113: “The plants were transferred into greenhouse in mid to late November and were self-pollinated in March of the following year to harvest F2 generation.” Please explain how the self-incompatibility was overcome.

Response: Since the self-incompatibility of Brasscia plants at the flowering stage, we performed hand pollination at the bud stage of F1 plants to overcome self-incompatibility. The relevant details were supplemented to the Materials and Methods section (Lines 121-124).

Page 3 lines 114-117: “From the F2 population of ‘Starbor F1’ (Figure 1a), plants with leaf scroll, medium height, and robust growth were chosen, and the F2 individuals from ‘Winterbor F1’ with high growth, compact plants, and vigorous growth were chosen (Figure 1b) as donor plants for IMC.” Please explain the difference between high growth, vigorous growth and robust growth as it is unclear.

Response: We revised and supplemented the specific selection criteria of F2 donor plants of IMC in more detail and clearly, as follows: “From the F2 population of ‘Starbor F1’ and ‘Winterbor F1’ (Figure 1), the F2 plants with phenotype characteristics as similar as possible to their respective F1 commercial variety were selected as donor plants for IMC. Specifically, the ‘Starbor F2’ and the ‘Winterbor F2’ plants exhibiting vigorous growth with compact multiple branches, dark green leaves, curled leaf margins, and free from disease and wilting were chosen. The overall height of ‘Winterbor F2’ plants was relatively slightly higher the ‘Starbor F2’ plants.” (Lines 126-132).

Page 3 line 130: “Mercury chloride exposure induces DNA damage, reduces fertility, and alters somatic and germline cells in Drosophila melanogaster”. (Mojica-Vázquez et al. 2019). It is questionable whether cells remained unaltered after these treatments.

Response: Disinfection of flower buds with a low concentration of 0.1% mercuric chloride in the plant microspore culture protocol is a commonly used method that does not adversely affect the microspore cells. The duration of disinfection ranged from 5 to 10 minutes due to differences in bud size among species. Unlike animal cell, there were thick cell walls from the flower bud epidermis to the anther epidermis, and the pollen wall on the surface of the microspore to protect the cells; Also, after mercuric chloride disinfection, the buds were washed three times with sterile distilled water to make sure that residual mercuric chloride on the bud surface was removed; Finally, the buds were mashed in B5 medium to isolate the microspore cells.

Page 3 line 137: Please give the full term for the abbreviation “NLN”.

Response: The NLN refers to the NLN liquid medium supplemented with 13% (w/v) sucrose, which has been supplemented (Lines 155-156).

Page 3 line 125: 2.2. Isolation of microspores should be written in past tense not as lab method preparations.

Response: We revised the 2.2 section with the past tense as you suggested (Lines 144-164).

Page 5 line 192: What traits were measured for petiole.

Response: We measured the length and width of petiole for DH lines regenerated plants. These were added in line 213.

Table 1: Please add “Concentration of TDZ” in the second column.

Response: We added the ‘Concentration of TDZ’ in the second column of Table 1 (Line 250).

Table 5: Please add four rows after S9 and D9: Mean, Min, Max, standard deviation or CV.

Response: We added the value of Mean, Min, Max and SD to Table 5 (Line 354).

Reviewer 2 Report

General Observations:

The authors conducted research using the plant growth regulator Thidiazuron to promote microspore embryogenesis and plant regeneration in Kale. They used the hormone to improve the efficiency of regenerating two Kale cultivars through the tissue culture technique. The research is of great importance to the scientific community as this contributes to reducing breeding cycles and has the potential for producing diseased free planting materials. However, Authors should consider using simple sentences to avoid ambiguity. Also, the selection criteria of the two cultivars for the research are unclear. In the discussions section, efforts should be made to consider linking all previous studies to the obtained results, since this is an original article and not a review. Overall the research is well written and I congratulate the authors for their effort, the pictures were particularly useful.

Other concerns 

•  L14, L43, consider changing ´fitness crowds´ to fitness enthusiasts or fitness populace. The sentence L14-L18 could be simplified to make them read better. Avoid long sentences as the ideas could be confusing.

•   L20, specify the control used, positive or negative control.

• L36, delete ´once´. In L41-42, replace ´chock-full of vitamins´ with very high in vitamins. Also, change ´health food´ to health attributes.

• L45-47 consider revising to make it readable and citation is required. The statement is disputable since plant introduction is crucial for improving crop variability. Thus, it complements the development of new varieties and broadens genetic variance to rapidly establish heterotic pools.

• L48 ´The availability of homozygous parent lines is the prerequisite for utilizing hybrid vigour in breeding´. This is true for self-pollinated species and hence not appropriate for a cross-pollinated crop like Kale. Thus, unless double haploids, the opposite is the case here, since heterogenous parents give better hybrid vigour in cross-pollinated species. Kindly clarify.

• L51 do you mean genetic diversity instead of hybrid diversity?

• L68, L127, L360 L363, L81, L85, L127, L385, L413, L415, and L419. Kindly check the citation style to conform with the Journal's requirements. Eg. Zhang et al. [23] Not ….Zhang et al. (2008),  Zhao et al. (2022), Madi et al. (2014) etc.

•  L74, change ´medium components´ to media constituents. Also, consider revising L75-77 for readability.

•  L183, Consider, diploid plants were… not ´was´. In L184, what were the different intensities measured according to the isolated peaks? L198, consider, genotypes were,…. not ´was´

• L198-203 clarify if data was normalized/ transformed in any way before analysis. What statistical analysis was performed before means comparison using Duncan multiple range test?

• L206, note that the plural for a medium is media. So apply appropriately 

• L212-214, L302, kindly indicate the level of significance. Also Clarify this claim, regarding the donor genotype factor in this statement …¨These results demonstrated that the donor genotype is the primary factor that directly determined microspore embryogenesis¨.  

• L271-272, revise to improve comprehension.

• L342, Mature? not clear,  and in lL370-373 consider breaking the sentence for clarity. L375, change  ´..to more reasonable detect other factors´. to …reasonably detect other factors. L381, consider using...It has been reported that...

Author Response

To Reviewer 2:

Comments to the Author:

General Observations:

The authors conducted research using the plant growth regulator Thidiazuron to promote microspore embryogenesis and plant regeneration in Kale. They used the hormone to improve the efficiency of regenerating two Kale cultivars through the tissue culture technique. The research is of great importance to the scientific community as this contributes to reducing breeding cycles and has the potential for producing diseased free planting materials. However, Authors should consider using simple sentences to avoid ambiguity. Also, the selection criteria of the two cultivars for the research are unclear. In the discussions section, efforts should be made to consider linking all previous studies to the obtained results, since this is an original article and not a review.

Overall, the research is well written and I congratulate the authors for their effort, the pictures were particularly useful.

Response: We thank the Reviewer for raising these concerns. Your insightful comments help us improve the manuscript. According to your suggestion, we carefully rechecked and used simple descriptions to refine sentences to avoid ambiguity. We also supplemented the clearer selection criteria of the two cultivars for the IMC in the Materials and Methods section (L126-132). In addition, we modified the discussion section in the revised manuscript to link all previous studies to the obtained results (L388-396, L425-429, L460-463).

Other concerns:

  1. L14, L43, consider changing ´fitness crowds´ to fitness enthusiasts or fitness populace. The sentence L14-L18 could be simplified to make them read better. Avoid long sentences as the ideas could be confusing.

Response: We removed the relevant sentence in lines 14 and 43 that containing the phrase ‘fitness crowds’, which may not be considered scientific language. As you suggested, the sentence L14-L18 was simplified.

  1. L20, specify the control used, positive or negative control.

Response: The ‘control’ in L20 referred to a blank setting without TDZ treatment (0 mg/L TDZ) in IMC. It was supplemented in L20 and L174.

  1. L36, delete ´once´. In L41-42, replace ´chock-full of vitamins´ with very high in Also, change ´health food´ to health attributes.

Response: We deleted the ‘once’ in L36. The ‘chock-full of vitamins’ was replaced with ‘it contains more vitamins’, and the sentence was corrected as follows: “Because it contains more vitamins and dietary fiber than Brassica and other vegetables, it has gained great popularity as a ‘superfood’.” in L42-L44.

  1. L45-47 consider revising to make it readable and citation is required. The statement is disputable since plant introduction is crucial for improving crop variability. Thus, it complements the development of new varieties and broadens genetic variance to rapidly establish heterotic pools.

Response: Your insightful suggestions led us to conceive that targeted introduction and utilization is a key link in improving crop variability. We revised the original sentence to ‘The worldwide popularity of the edible leafy kale has led to a surge in demand, prompting breeders to develop landraces suitable for local specific conditions. ​Thus, the breeding of improved varieties and the cultivation of new varieties are of great significance to the edible leafy kale’ in L46-L49. The study from Šamec et al. [2] was cited.

  1. L48 ´The availability of homozygous parent lines is the prerequisite for utilizing hybrid vigour in breeding´. This is true for self-pollinated species and hence not appropriate for a cross-pollinated crop like Kale. Thus, unless double haploids, the opposite is the case here, since heterogenous parents give better hybrid vigour in cross-pollinated species. Kindly clarify.

Response: We thank the Reviewer for bringing this to our attention, and the incongruent statement was removed.

  1. L51 do you mean genetic diversity instead of hybrid diversity?

Response: Yes, we changed the ‘hybrid diversity’ to ‘genetic diversity’.

  1. L68, L127, L360 L363, L81, L85, L127, L385, L413, L415, and L419. Kindly check the citation style to conform with the Journal's requirements. Eg. Zhang et al. [23] Not ….Zhang et al. (2008),  Zhao et al. (2022), Madi et al. (2014) etc.

Response: We thank the Reviewer for pointing this out. We rechecked the citation style and corrected formats to conform with the Journal's requirements.

  1. L74, change ´medium components´ to media constituents. Also, consider revising L75-77 for readability.

Response: We changed ‘medium components’ to ‘media constituents (Line 79). The long sentence was split into ‘A key internal factor is the genotype of donor material. Under the same external conditions, plants with different genotypes in the same species may show varying abilities in microspore embryogenesis.’ in L80-82.

  1. L183, Consider, diploid plants were… not ´was´. In L184, what were the different intensities measured according to the isolated peaks? L198, consider, genotypes were,…. not ´was´

Response: The singular and plural forms in L203 and L218 were corrected. In L204, the intensities referred to the relative fluorescence intensity emitted by the DNA dye PI after absorbing laser excitation. The DNA absorption peak of the diploid was at approximately the 200 on the x-axis, while samples with peaks at the 100 and 400 channels were haploid and tetraploid, respectively.

  1. L198-203 clarify if data was normalized/ transformed in any way before analysis. What statistical analysis was performed before means comparison using Duncan multiple range test?

Response: â€‹The data was not transformed in any way prior to the analyses. The experimental data was analyzed using analysis of variance (ANOVA) with SPSS 26.0 software, and the means comparison analysis was done using Duncan’s multiple range test (DMRT) (L221-222).

  1. L206, note that the plural for a medium is media. So apply appropriately 

Response: We thank the Reviewer for pointing this out. We checked the use of the singular and plural of this word in the manuscript and made changes in L227, L239, L297 and L298.

  1. L212-214, L302, kindly indicate the level of significance. Also Clarify this claim, regarding the donor genotype factor in this statement …¨These results demonstrated that the donor genotype is the primary factor that directly determined microspore embryogenesis¨.  

Response: In L233-235, to be more rigorous, we replaced the ‘significantly’ with the ‘obviously’, and the statement about the donor genotype factor, we revised the sentence to ‘These results suggest that the donor genotype is a key internal determinant of microspore embryogenesis’; In L324, it was revised to ‘there was no obvious difference in doubling efficiency between the two genotypes of edible curly kale’.

  1. L271-272, revise to improve comprehension.

Response: In L291-292, we revised the ‘To address this, the ‘Starbor F2’ seedlings, mostly of which failed to take root, therefore, were selected as the test material for investigating the rooting rate’ to “To address this issue, we selected ‘Starbor F2’ seedlings, most of which failed to take root, as the test material for the rooting rate”.

  1. L342, Mature? not clear,  and in lL370-373 consider breaking the sentence for clarity. L375, change  ´..to more reasonable detect other factors´. to …reasonably detect other factors. L381, consider using...It has been reported that...

Response: In L364, the ‘It has been considerably mature’ was revised to ‘It has made great progress’. In L400-403, the sentence was revised to ‘For this reason, we selected phenotypically superior plants from the F2 generations of the two genotypes of curly kale for IMC. It could allow us to obtain a relatively high microspore embryogenesis rate and to cultivate abundant DH lines from them to meet our breeding objectives.’ In L408 and L415, we modified these two places as you suggested.

Reviewer 3 Report

I conducted a thorough review of Thidiazuron promoted microspore embryogenesis and plant regeneration in curly kale (Brassica oleracea L. convar. acephala var. sabellica). The paper clearly shows the protocol for in vitro embryogenesis of Brasica oleacea species. I found the manuscript very interesting because the whole protocol of successful plant regeneration by embryogenesis was presented. Also, the manuscript is well-written and overall easy to follow. However, the paper needs to be improved. If the authors provide a revised version with some minor corrections the article will be acceptable for publication.

Line 282 Did you have control medium (without hormones). Please add it in material and methods, and in results. 

Author Response

To Reviewer 3:

Comments to the Author:

I conducted a thorough review of Thidiazuron promoted microspore embryogenesis and plant regeneration in curly kale (Brassica oleracea L. convar. acephala var. sabellica). The paper clearly shows the protocol for in vitro embryogenesis of Brasica oleacea species. I found the manuscript very interesting because the whole protocol of successful plant regeneration by embryogenesis was presented. Also, the manuscript is well-written and overall easy to follow. However, the paper needs to be improved. If the authors provide a revised version with some minor corrections the article will be acceptable for publication.

  1. Line 282 Did you have control medium (without hormones). Please add it in material and methods, and in results. 

Response: We thank the Reviewer for raising this concern. Through our previous investigations, the 0.1 mg/L NAA added to MS medium with optimum rooting efficiency is served as a common formulation for promoting the rooting of regenerated plants and, therefore, it was used as the control in this study. To promote rooting of the two genotypes of curly kale, we tested the effect of 1/2 MS media vs. MS media by applying two other increasing NAA concentrations (0.2 and 0.3 mg/L). We added details and cited relevant studies in the Materials and Methods and Results for clarity (L191-192 and L294-295).
